# Form-Dependent Toxicity of Silver Nanomaterials in Rainbow Trout Gills

**DOI:** 10.3390/nano13081356

**Published:** 2023-04-13

**Authors:** Joëlle Auclair, Patrice Turcotte, Christian Gagnon, Caroline Peyrot, Kevin James Wilkinson, François Gagné

**Affiliations:** 1Aquatic Contaminants Research Division, Environment and Climate Change Canada, Montreal, QC H2Y 2E7, Canada; joelle.auclair@ec.gc.ca (J.A.);; 2Chemistry Department, Montréal University, Montreal, QC H3C 3J7, Canada

**Keywords:** silver nanoparticle, form, availability, dissolution, oxidative stress, lipids, genotoxicity

## Abstract

The toxicity of the form of nanoparticles is presently not well understood. The purpose of this study consists in comparing the toxicity of various forms of silver nanoparticles (nAg) in juvenile rainbow trout *Oncorhynchus mykiss*. Juveniles were exposed to various forms of polyvinyl-coated nAg of similar size for 96 h at 15 °C. After the exposure period, the gills were isolated and analyzed for Ag uptake/distribution, oxidative stress, glucose metabolism, and genotoxicity. Higher levels of Ag were detected in gills in fish exposed to dissolved Ag followed by spherical, cubic, and prismatic nAg. Size-exclusion chromatography of gill fractions revealed that the dissolution of nAg was observed for all forms of nAg where prismatic nAg released more important levels of Ag in the protein pool as in fish exposed to dissolved Ag as well. The aggregation of nAg was more important for cubic nAg in respect of the other forms of nAg. The data revealed that lipid peroxidation was closely associated with protein aggregation and viscosity. Biomarkers revealed changes in lipid/oxidative stress and genotoxicity, which were related to the loss of protein aggregation and inflammation (NO_2_ levels), respectively. In general, the observed effects were found for all forms of nAg where the effects from prismatic nAg were generally higher than for spherical and cubic nAg. The strong relationship between genotoxicity and inflammation response suggests the participation of the immune system in the observed responses of juvenile fish gills.

## 1. Introduction

Silver nanoparticles (nAg) represent an important part of the nanotechnology industry because of their antimicrobial properties [1]. They are found in many consumer and medical products such as clothes (shirts, pants, socks etc.), tubing, and various devices as aseptic agents and odor prevention [2]. These applications resulted in increased contamination of various aquatic ecosystems from the inadvertent release through solid waste leaching and municipal/industrial wastewaters [3]. Municipal wastewater is an important source of nAg in the aquatic environment where they are found mostly in suspended matter and sludges [4]. While most of the nAg are removed from the effluents (95%), some nAg remain in the effluent at the low ng/L range [5]. Silver nanoparticles come in many sizes (1–100 nm), coatings (citrate, polyvinylpyrrolidone, silicate), and forms (sphere, cube, plate/prism, and tube), which can influence the water–solid partition, half-lives, and reactivity [6]. Transmission electron microscopic observation of aged uncoated spherical and cubic nAg in marine and freshwater revealed that cubic nAg were eroded around the edges and that spherical nAg formed large aggregates. In another study, the catalytic activity of nAg wires, prisms, cubes, and spheres toward the plasmon-resonance-induced oxidation of p-aminothiophenol differed in the following: triangular prisms ~ spheres > wires cubes [7]. The geometry of nanoparticles could influence the thickness and composition of the protein corona at the surface, a process usually occurring in both the extra- and intracellular environments, and could change the resulting toxicity of these nanoparticles. In *Staphylococcus aureus*, prismatic nAg was more toxic than spherical, tubular, and cubic nAg were [8]. The thickness of fetal bovine serum proteins on the nanoparticles was lower for prismatic and tubular nAg compared to spherical and cubic nAg, with the spherical nAg containing six times more albumin at the surface. Although tubular nAg had low albumin content, the lower toxicity was explained by very long dimensions (5 µm) of the wires and a reduced availability for bacteria. Protein corona thickness was lower in the most toxic forms of nAg, indicating that a tenuous protein shell thickness at the surface of nAg could favor dissolution more quickly. In addition, the amount of low-molecular-weight proteins tended to be higher in steep-edged nanostructures such as prismatic and cubic nAg. This suggests that cubes and prisms have more surface tensions/constraints as shown by reduced protein thickness (for large proteins) and an increased amount of low-molecular-weight proteins.

The toxicity of various shapes of polyvinylpyrrolidone-coated nAg at the same size range was recently examined in freshwater organisms [9]. The study revealed that the form of nAg produced different effects, not always attributable to ionic Ag. The formation of liquid crystals in cells was increased as follows: prismatic > cubic > spherical nAg, suggesting again that steep-edged nAg produce more steric tensions and could disrupt the physical properties in cells (i.e., liquid crystal formation and viscosity). It is well known that the oxidative properties of ionic Ag released from nAg could lead to oxidative stress and damage such as lipid peroxidation (LPO) and DNA damage at relatively low exposure concentrations between 10 and 100 µg/L in Nile tilapia [10]. In respect of genotoxicity, the exposure of Nile tilapia to nAg increased DNA strand breaks in liver cells within the first 24 h and decreased afterward, suggesting effectiveness in DNA repair mechanisms [11]. Nevertheless, cytogenetic damage (irreversible) was observed in peripheral blood leucocytes as shown by nuclear abnormalities and micronuclei formation. Moreover, these studies were conducted with ovoid-shaped nAg, but other forms were not investigated in these studies. In this respect, the toxicity of various forms of nAg could result not only from the release of toxic Ag ions but from uptake differences, stability in the cell’s environment, and crowding effects from the various geometries of nAg. In this study, a novel size-exclusion chromatography methodology [12] was used to better understand the stability (aggregation vs. dissolution of Ag) of various forms of nAg in the intracellular environment.

The purpose of this study was therefore to compare the toxic effects of various forms of PVP-coated nAg of similar dimensions in juvenile rainbow trout in an attempt to address the null hypothesis of this study: the geometry of nAg has no influence toward toxicity. The nAg were all coated with polyvinylpyrrolidone and were of a similar size range (50–70 nm) to optimize form-related effects. As gills are the first target tissues exposed to suspended matter and dissolved components in surface water, the accumulation of Ag in gill tissues and the intracellular distribution of Ag were examined by low-pressure size-exclusion chromatography. The toxic effects were investigated by following the changes in viscosity, protein aggregation, oxidative stress, DNA damage, and lipid and glucose metabolisms in juvenile fish gills. An attempt was made to identify critical events leading to toxicity in fish gill tissues.

## 2. Materials and Methods

### 2.1. Exposure of Rainbow Trout Juveniles

Rainbow trout juveniles (length 4 ± 0.2 cm; weight: 3 ± 0.3 g) were maintained for one month at 15 °C using a 16–8 h light and dark cycle with constant aeration according to a standardized method [13]. The aquarium consisted of UV-treated and charcoal-filtered tap water of the city of Montréal (conductivity: 280 µScm^−1^, total organic carbon: 1–3 mg/L, total suspended solids < 1 mg/L). Juveniles were exposed to 5 and 50 µg/L of total Ag as polyvinylpyrrolidone (PVP)-coated silver nanoparticles (nAg) of a similar size range (70–80 nm) but of different form: sphere, cube, and prism/platelet (NanoXact, NanoComposix Inc., San Diego, CA, USA). Prismatic nAg was a triangular prism with rounded edges with a thickness between 5 and 10 nm. The characterization of the 3 forms of nAg was provided in a previous study [9]. Dynamic light scattering and zeta potential analyses revealed that the mean sizes of the nanoparticles were 73, 70, and 86 nm for the spherical, cubic, and prismatic nAg, respectively. The zeta potential of spherical and cubic PVP-coated nAg was −38 and −40 mV, respectively. The surface area was 31,416, 60,000, and 6000 nm^2^ for nAg spheres, cubes, and prisms, respectively. Fish were also exposed to 5 µg/L of dissolved Ag (AgNO_3_) for comparison purposes. These concentrations were selected based on the upper range for total Ag found in municipal effluents [14,15]. For each nAg and dissolved Ag, stock solutions (1 mg/L) were prepared on the day of exposure in MilliQ water to prevent aggregation based on previous studies [16]. Silver concentrations were prepared at 5 and 50 µg/L as the total Ag for each nanoform of Ag in 60 L containers lined with polyethylene bags at 15 °C with aeration. The negative control consisted of tap water only. Water pH, conductivity, and dissolved oxygen were measured daily and monitored for any signs of distress, anormal behavior, and mortality. At the end of the exposure time (96 h), the fish were anesthetized in 0.1% MS222 buffered to pH 7.4 with 1 mM NaHCO_3_ (Sigma-Aldrich, Oakville, ON, Canada) following the ethical committee guidelines for aquatic vertebrates. The length and weight of fish were taken, and gills were quickly removed and homogenized in ice-cold 250 mM sucrose containing 20 mM Tris-acetate, pH 7.5, 1 µg/mL of apoprotinin B (protease inhibitor), and 0.1 mM EDTA. Homogenization was achieved using a Polytron tissue grinder at a 30 s burst (low speed setting) on ice to prevent heating of the sample. The homogenates were centrifuged at 3000× *g* for 10 min and the supernatant was stored at −85 °C until biomarker analysis.

### 2.2. Stability of Ag in the Exposure Media and Detection Gill Tissues

Preliminary experiments were performed to examine the stability of the nAg form and dissolved Ag in the exposure media. Water samples were taken at 1 h and 96 h in the exposure media (without fish) and the dissolved levels of Ag (following filtration at 0.45 µm on a nanocellulose membrane) were determined using ion-coupled plasma quadrupole mass spectrometry as previously described [17]. The water samples (10 mL) were acidified in 1% HNO_3_ prior to analysis. The relative levels of Ag in gills and in the chromatographic fractions (see below) were determined using a chemiluminescent (CL) methodology [18]. Briefly, the supernatant (50 µL) was mixed with 200 µL of alkaline 0.1 M Na_2_CO_3_ containing 0.5 mM luminol (freshly prepared) and luminescence readings were integrated for 1 min. Positive controls were prepared with dissolved Ag (0.5 µM) and nAg (2 µM as Ag) for calibration. Dissolved Ag emitted 8–10 times more light than an equimolar concentration of Ag such as nAg. The chemiluminescence (CL) data were expressed as µmole Ag/mg proteins.

### 2.3. Size-Exclusion Chromatography

The distributions of Ag in the gill homogenate fractions were analyzed using a recent methodology for nanoparticles’ separation by low-pressure size-exclusion chromatography [12]. Briefly, a 40 × 1 cm column was prepared with Sephacryl S500 (capable of resolving nanoparticles from the protein pools based on size) using 1 mM KH_2_PO_4_, pH 7.4, containing 0.2% Tween-20 (non-ionic detergent) as the elution buffer. The elution buffer was found to prevent nanoparticle aggregation, the adsorption of proteins, and solubilization of any lipid structures (vesicles) present in the sample. The gill homogenates were centrifuged at 3000× *g* for 10 min, and a 100 µL sample of the supernatant was added to 100 µL of 0.2% Tween-20 and was injected directly to the column. The column was eluted at a flow rate of 0.75 mL/min and 1 mL fractions were collected after passing through 280 nm UV and conductivity detectors. The collected fractions were analyzed for Ag using the alkaline luminol CL-assay as described above [18] and at 410 nm for nAg detection using plasmon resonance spectroscopy [19]. The total volume of the column (Vt) was determined with 1 µg/mL of NaCl (conductivity), 25 nm citrate-coated nAg spherical and polystyrene fluorescently labeled nanoparticles (50 and 100 nm diameter), serum albumin (67 kdA) and rat metallothionein (6–7 kDa) for the protein pool range, and NaCl for low-molecular-weight salts. The void volume was estimated at 7 mL (Ve/Vt = 0.21) representing the upper range limit of the column estimated at 140 nm diameter particles. The spherical nAg eluted at a volume of 20 mL (Ve/Vt = 0.6) with some presence of aggregates at Ve/Vt = 0.21–0.25. The elution data were normalized to 1 to facilitate visual changes between chromatograms.

### 2.4. Biochemical Assessments

Rainbow trout (*n* = 8) were analyzed for fork length, body, and gill mass. The following endpoints were performed using compact, hand-held instruments for fluorescence (TBS-380, Turner Biosystem, Sunnyvale, CA, USA) and colorimetric analyzers on dried pads embedded with the appropriate reagents (EMP, URS-11 Solid phase pads, Wenzhou, China). The pads were purchased commercially as test strips for chemical analysis in biological samples (EMP, test strips, Wenzhou, China). The gill homogenate fraction (S3) was diluted with 2 volumes of MilliQ water and were analyzed for soluble proteins, bilirubin (Bil; heme degradation), glucose, acetoacetate (ketone body), nitrite (NO_2_), peroxidase, esterase activities, and antioxidant capacity. Esterase activity was determined by blotting 100 µL of the S3 fraction to a solid phase pad containing 4 mg of N-toluene sulfonyl alanine ester/g pad as described by the manufacturer (EMP, URS-11 reader). Peroxidase (Perox) activity was determined using cumene hydroperoxide (6 mg/g) and tetramethylbenzidine reagent (40 mg/g pad) as described above. Total soluble proteins were determined using a pad embedded with 3 mg of tetrabromophenol blue/g pad and read by the hand-held colorimetric analyzer. The levels of acetoacetate (a ketone body) were determined using 77 mg of nitroprusside/g pad and analyzed by the reader. The levels of glucose were determined using a pad embedded with 0.16 mg of glucose oxidase/g pad containing 70 mg of KI/g pad and 6 mg of peroxidase and analyzed by the reader as above. The levels of bilirubin (Bil) were determined using 0.5% 2,4-dichloroanilide diazonium/g pad in 1 mg of KH_2_PO_4_ buffer at pH 7.4/g pad. The antioxidant potential was determined as ascorbic acid equivalents using a pad embedded with 3 mg of 2,5-dichlorophenolindophenol buffered at pH 7.4 with 1 mg of KH_2_PO_4_ at pH 7.4/g pad and analyzed by the colorimeter as above. The levels of nitrites (NO_2_) were determined using a pad embedded with 45 mg of p-arsanilic acid and 0.1 mg of 3-hydroxy-1,2,3,4 tetrahydrobenzo-(h)-quinolin/g pad. All solid phase readings were validated with standard additions of each chemical or with varying amounts of S3 fraction for enzyme assessments (esterase and peroxidase), and solid phase readings were normalized to protein contents in the homogenate fraction as determined above. Lipid peroxidation (LPO) and DNA damage were determined by the malonaldehyde (thiobarbituric acid reactants) and DNA strand assays using fluorometry as previously described [20,21]. The levels of LPO were expressed as thiobarbituric acid reactants (TBARS) fluorescence units determined at 485 nm excitation and 540 nm emission (TBS-380, Turner Biosystems, Sunnyvale, CA, USA) with constant instrument calibration with the internal standard (fluorescein). The levels of DNA strand breaks were determined on the supernatant containing small, protein-free DNA strands from the precipitated genomic protein-rich DNA fraction using fluorometry at 350 nm excitation and 460 nm emission (TBS-380, Turner Biosystems, Sunnyvale, CA, USA) using the Hoescht dye. The dye was prepared in Tris-base (0.1 M) in high salt (0.4 M NaCl) and 4 mM deoxycholate detergent to prevent interferences from SDS and high pH during fluorescence readings [22,23]. The instrument was calibrated with quinine sulfate internal standard and 1 µg/mL of salmon sperm DNA. The data were expressed as DNA equivalents/mg proteins. Total lipids were determined using the Nile Red methodology [24]. Briefly, a 50 µL volume of the S3 fraction was mixed with 200 µL of 10 µM Nile Red in MilliQ water and fluorescence readings were taken at 485 nm excitation and >540 nm emission (TBS-380, Turner Biosystems, Sunnyvale, CA, USA). Standard solutions of triton X-100 (10 mg/mL) were used as positive controls and the instrument was calibrated with the fluorescein internal standard. The data were expressed as fluorescence units/mg proteins.

## 3. Data Analysis

Trout (*n* = 8) were exposed in 60 L containers for each of the 8 treatment groups (controls, 5, and 50 µg/L for spherical, cubic, and prismatic PVP-nAg, and 5 µg/L for dissolved Ag) and the exposure experiments were repeated twice. The data were analyzed using the non-parametric analysis of variance with the Conover–Inman test as the multiple comparison *post hoc* test to highlight differences between treatment groups. Correlation analysis was performed using the Pearson-moment procedure followed by principal components to identify the most important biomarkers explaining the data variance and interrelationships between them. Significance was set at *p* ≤ 0.05 and all tests were performed using the SYSTAT software package (version 13).

## 4. Results

Rainbow trout juveniles were exposed to two concentrations (5 and 50 µg/L) of different forms of nAg (sphere, cube, and prism/plate) and to 5 µg/L for dissolved Ag for 96 h at 15 °C. The stability of these suspensions was determined in the aquarium water after 1 and 96 h of dissolution time (Table 1). Dissolved Ag remained stable over the exposure time. The forms of nAg were fairly stable (60–70% of Ag remaining after 96 h) in exposure water with relatively low changes in a decreasing manner: prismatic > cubic and spherical nAg. The relative levels of Ag were determined using a chemiluminescence assay that can detect both nAg and dissolved Ag in gill tissues (Figure 1). As expected, the levels of Ag were highest in fish exposed to the dissolved Ag form and were 3 times higher than the reported Ag concentrations from the nAg forms at the same concentration (5 µg/L). Spherical nAg was significantly increased from controls for the 50 µg/L exposure group. The relative levels of Ag were significantly increased from controls for each exposure concentration (5 and 50 µg/L) group for cubic and prismatic forms of nAg. Morphometric changes in fish were determined by following the condition factor (fish weight/fork length), hepatosomatic index (HSI), and the gill index (Table 2). Only the HSI was significantly affected by some forms of nAg: at 5 and 50 µg/L for spherical nAg and at 5 µg/L for prismatic nAg and dissolved Ag. The gill homogenate supernatant was further resolved by size-exclusion chromatography to gain some insights into the stability of nAg in tissues and the distribution of Ag in the cellular protein pool (Figure 2). In the following chromatograms, the proteins were analyzed by the absorbance at 280 nm, the nanoparticles were analyzed at 410 (plasmon resonance) nm, and Ag was analyzed by the chemiluminescent methodology. The total volume (Vt) was estimated with NaCl (conductivity), giving Ve/Vt = 1, and the void volume was evaluated with Ve/Vt = 0.25. For spherical nAg, the presence of nAg eluted at Ve/Vt = 0.4 corresponded to a molecular size of 40–60 nm based on the column calibration with 100, 50, and 20 nm polystyrene nanobeads (Figure 2A). Two other Ag peaks were found, corresponding to Ve/Vt = 0.62 (corresponding to 10–20 nm size) and Ve/Vt = 0.8–0.9, corresponding to low-molecular-weight proteins and ligands such as glutathione and amino acids. Indeed, the protein pools eluted at Ve/Vt = 0.6–0.95 were dissolved components (KCl) eluted at the total column volume of Ve/Vt = 1. Metallothionein, a silver binding protein with 7000 d mass, eluted at Ve/Vt = 0.75, suggesting that Ag was not only sequestered by metallothioneins but also with other low-molecular-weight proteins based on the UV absorption at 280 nm. For cubic nAg, a similar pattern was found to spherical nAg, but a peak (410 nm and Ag) was found at the void volume at Ve/Vt = 0.25–0.35, suggesting large aggregates > 200 nm (Figure 2B). For prismatic nAg, a higher proportion of Ag was found in the low-molecular-weight range (Ve/Vt = 0.7–1) and was observed with no evidence of large aggregates. Based on the absorbance at 280 nm, a significant number of proteins was associated with prismatic nAg, while little to no proteins were associated with spherical and cubic nAg. For dissolved Ag, most of the Ag was in the low-molecular-size range overlapping the proteins (A280 nm), suggesting that ionic Ag was mostly bound to protein pools.

The energy status of juvenile rainbow trout was determined by following the glucose, acetoacetate, and esterase (Est) activity and total lipids in gill tissues (Figure 3). Glucose and acetoacetate levels were not affected by the forms of Ag but Est activity was increased by spherical nAg at 50 µg/L. Lipids were increased by all the forms of nAg but not with dissolved Ag. Correlation analysis revealed that Est activity was significantly correlated with the relative Ag levels in gills (r = 0.32). Total lipids were negatively correlated with glucose levels (r = −0.30). Oxidative stress (inflammation) and damage were determined by following the changes in peroxidase, lipid peroxidation, NO_2_ levels, and the antioxidant potential in gills from juvenile rainbow trout exposed to the various forms of Ag (Figure 4). Peroxidase activity was significantly increased by dissolved Ag, and only a marginal increase was observed with prismatic nAg. Correlation analysis revealed that peroxidase activity was significantly correlated with acetoacetate levels (r = 0.49). The levels of LPO tended to be higher in the exposure group but was only significantly higher for spherical nAg at 50 µg/L. The correlation analysis of LPO was significantly related to total lipids (r = 0.39). Globally, the NO_2_ levels were also higher with all forms of nAg including dissolved Ag but were significant with dissolved and spherical nAg. Correlation analysis revealed that NO_2_ levels were significantly correlated with peroxidase activity (r = 0.61), acetoacetate (r = 0.54), and glucose (r = 0.43). The antioxidant potential was also significantly decreased by all forms of nAg and Ag. The correlation revealed that the antioxidant potential was significantly correlated with NO_2_ levels (r = −0.33), peroxidase activity (r = −0.31), and glucose levels (r = 0.31).

Tissue damage in trout exposed to various forms of Ag was determined by following the levels of DNA breaks, protein aggregation, and heme degradation product bilirubin (Figure 5). Viscosity was also included although not strictly a marker of damage. DNA damage was present in all Ag forms where the genotoxic potential of dissolved Ag is similar to all forms of nAg. Correlation analysis of DNA damage revealed a significant trend with NO_2_ levels (r = 0.68), peroxidase activity (r = 0.45), glucose (r = 0.87), and acetoacetate (r = 0.39). Protein aggregation was significantly reduced by cubic and spherical nAg for the low and high concentrations. A reduction in protein aggregation also occurred for spherical nAg but only at high concentrations. Correlation analysis revealed significant relationships with glucose (r = 0.36) and LPO (r = 0.46). The levels of Bil were determined as an indicator of heme damage. Bil levels were decreased in fish exposed to the lowest concentration of cubic nAg and at the highest concentration of prismatic nAg. Correlation analysis revealed that Bil levels were correlated with glucose (r = 0.62), antioxidant potential (r = 0.67), and protein aggregation (r = 0.3). Viscosity changes tended to be lower by the forms of nAg but not significantly. However, a significant increase in viscosity was observed for the low concentration of spherical nAg. The correlation analysis of viscosity revealed a significant correlation with esterase activity (r = −0.36), LPO (r = 0.72), protein aggregation (r = 0.63), and total lipids (r = 0.37). In an attempt to gain a global view of the various responses, a principal component analysis was performed (Figure 6). The total variance was 50% and the following biomarkers had the highest component weight (>0.7): viscosity, glucose, DNA proteins, NO_2_ levels, and gill protein levels. Changes in viscosity were closely associated with protein aggregation and LPO. DNA damage and NO_2_ levels were closely related with each other, peroxidase activity, and bilirubin, suggesting the involvement of oxidative stress and increased hemoprotein turnover. Gill Ag levels were closely related with acetoacetate and esterase activity, suggesting impacts on gluconeogenesis and lipid mobilization.

## 5. Discussion

The fish accumulated 3–5 times more Ag in fish exposed to ionic/dissolved Ag compared to the nAg forms. At low concentrations, cubic and prismatic nAg accumulated more in gills compared to spherical nAg. A possible explanation for this could be that a more reactive form of Ag is released from cubic and prismatic nAg than for spherical nAg in tissues. Indeed, the CL assay used in the present study could detect both nanoparticulate and dissolved Ag but was reported eight times more sensitive with the latter [18]. The same accumulation rate (threefold accumulation in gills) of *Oreochromis mossambicus* fish was exposed to ionic compared to spherical nAg [10]. The half-lives of the selected forms of nAg were within the order of ≥114 h with an estimated time for 126, 114, and 150 h for spherical, cubic, and prismatic nAg, respectively. This suggests that fish were readily exposed to suspensions of nAg during the exposure period of 96 h and could pose a risk to predators feeding on fish (including humans) in the vicinity of urban pollution. This is in keeping with a previous study that examined the stability of citrate-coated nAg in various waters and wastewaters [25]. The half-life of spherical nAg was of the order of 100 days in untreated wastewaters and the proportion of ionic Ag+ did not increase over time. This study also showed that the size distribution of nAg was fairly stable during that time and the free Ag+ that occurred represented 40–50% of the total Ag levels at time 0 where this proportion remained fairly constant in time in untreated water samples. This is consistent with the previous observation where the levels of suspended Ag were significantly correlated with nAg in treated municipal effluents [5]. In some conditions, the nanoparticles were more quickly degraded in treated municipal effluents, synthetic effluent, and reclaimed waters with nAg half-lives < 10 h. This suggests that nAg could be relatively stable in complex media (untreated wastewaters) but could quickly dissolve in treated effluents and reclaimed waters. The coating of the nAg forms was PVP, which confers more stability than citrate, which is electrostatic in nature (i.e., exchangeable with other anions). Indeed, it was found that PVP-coated nAg did not aggregate in water, even at an ionic strength equivalent to 10 mM CaCl_2_, compared to citrate-coated nAg [26]. In this study, we examined the stability of nAg in gill tissue extracts by size-exclusion chromatography where Ag was found in the low-molecular-weight range (Figure 2). For spherical and cubic nAg, about 50% of Ag was found in the low-molecular-weight range, and this proportion rose to about 70% for prismatic nAg, suggesting increased degradation in cells. Detectable amounts of proteins (A280 nm) were present in the size range of prismatic nAg, suggesting the adsorption of proteins to the nanoparticles. The in vitro exposure of human mesenchymal stem cells and human keratinocytes to PVP-coated nAg spheres and prisms lost more than 90% of their volume after 24 h at 37 °C [27]. It was found that the cell accumulation of prismatic nAg was facilitated by membrane fluidity in mesenchymal stem cells compared to keratinocytes. In this study, the edges of prismatic (platelet) nAg became quickly eroded, showing more surface tension favoring the liberation of silver to the low-molecular-weight pool in keeping with the increased dissolution of prismatic nAg compared to spherical and cubic PVP-coated nAg in the present study. Protein corona formation at the surface of metallic nanoparticles in cells is a common occurrence [3,28]. The liberation of Ag from interactions with metallothioneins, a thiol-rich metal binding proteins with high affinity for Ag, led to nAg dissolution [28]. The formation of the protein corona with a non-metal binding protein ceruloplasmin led to a more stable corona–nanoparticle complex. Based on the chromatographic elution profiles of the various forms, Ag was detected in fractions consistent with metallothioneins (Ve/Vt = 0.75–0.8) with the prismatic nAg and dissolved Ag in fish gills. It is noteworthy that metallothioneins are devoid of aromatic amino acids and therefore do not absorb at 280 nm. The toxicity of prismatic (plate) nAg was higher than those of spherical and tubular nAg in the fish cell line and zebrafish embryos [29]. It was suggested that the toxicity of nAg plates was attributed to surface roughness/heterogenicity (point defects), and treatment of nAg plates with cysteine reduced the toxicity and oxidative stress induced by the nanoparticle and dissolved Ag.

The toxicity of the forms of PVP-coated nAg was coated in freshwater unionids and revealed the following order of toxicity: prism < cube < sphere based on protein-ubiquitin (damage) levels [9]. In another study, nAg prisms reduced the air survival time in dreissenids compared to PVP-coated spherical and cubic nAg [16]. Reduced acetylcholinesterase activity was somewhat reduced more strongly with nAg plates (prisms). Prismatic and spherical nAg reduced growth and reproduction in *Caenorhaditis elegans* than tubular/wire nAg [30]. In keratinocytes, truncated prismatic nAg was more toxic that the spheric nanoparticles [31]. This suggests that asymmetric and elongated structures (cylinders, rectangles, etc.) contribute to more surface tension, favoring toxicity in part at least from the liberation of dissolved components. At some point, the toxicity could be reduced when dimensions tend to wire-like geometries, limiting absorbance and cell internalization. The toxicity of prismatic nAg was also more toxic than spherical nAg in zebra fish embryos [32]. Toxicity was not necessarily related to the release of ionic Ag and the particle form contributed to toxicity as well.

It is noteworthy that genotoxicity was one of the principal components associated with each form of Ag. The extent of DNA damage was associated, in part at least, to Ag levels in gills (r = 0.46), suggesting that internalization in cells is a contributing factor to effects. The levels of NO_2_, an indicator of inflammation from the formation of peroxynitrite in phagosomes and the activation of the NO_2_ pathway [33], were highly correlated with DNA strand breaks (r = 0.96), suggesting the contribution of the immune function in fish exposed to the forms of Ag. Increased NO_2_ levels could form 3-nitrotyrosine in proteins and contribute to protein aggregation and viscosity. This was supported by the significant correlations between NO_2_ levels and viscosity (r = 0.88) and protein aggregation (r = 0.65). The addition of NO_2_ groups on tyrosine increases the polarity of the tyrosine, leading to protein aggregation and viscosity. The genotoxicity of nAg was reported in fish in the liver and muscle at the level of DNA breaks and the formation of micronuclei occurring after 48 h [11]. The genotoxicity was higher in citrate-coated spherical nAg compared to PVP-coated nAg in rainbow trout [34], suggesting the involvement of surface interactions (citrate coating involves electrostatic interactions while PVP coating involves encapsulation by neutral PVP) and perhaps dissolution of nAg from citrate coated nAg. Encapsulated PVP-coated nAg could interact at the surface of the cell membrane and enter by pinocytosis and fuse with lysosomes/phagosomes, leading to peroxynitrite and NO_2_ production, hence the strong correlation with NO_2_ levels. This was supported by transmission electron microscopy of gilthead seabream head-kidney leucocytes showing phagocytosis and pinocytosis of silver nanospheres [35]. Significant reductions in leucocyte viability were observed at concentrations between 1 and 10 µg/mL in addition to reduced phagocytosis activity and respiratory burst. The presence of nAg in cells was confirmed by transmission electron microscopy following pinocytosis and phagocytosis.

In conclusion, exposure to selected forms of nAg led to an accumulation of Ag in gills but to a lower degree than ionic Ag. Size-exclusion chromatography data revealed some degree of dissolution of Ag to the low-molecular-weight range and was highest with prismatic nAg compared to spherical and cubic nAg. Cubic nAg showed a higher aggregation as revealed by increased nAg at the void volume of the column. Effect biomarkers revealed that forms of nAg induced important changes in glucose levels, LPO, and genotoxicity. Oxidative stress (LPO) was closely associated with decreased protein aggregation and viscosity, while DNA damage was strongly related to NO_2_ levels, suggesting that oxidative stress was mediated by inflammation. Although toxicity involved the dissolution of Ag from the various forms of nAg into the low-molecular-weight-range protein/peptide pool, the toxicity was higher for prismatic nAg, leading to stronger effects toward lipid levels, oxidative stress, and DNA damage. This suggests that the release of more complex forms of nAg in respect of the usual spheres could produce more impacts in fish in respect of the release of dissolved Ag in tissues.

## Figures and Tables

**Figure 1 nanomaterials-13-01356-f001:**
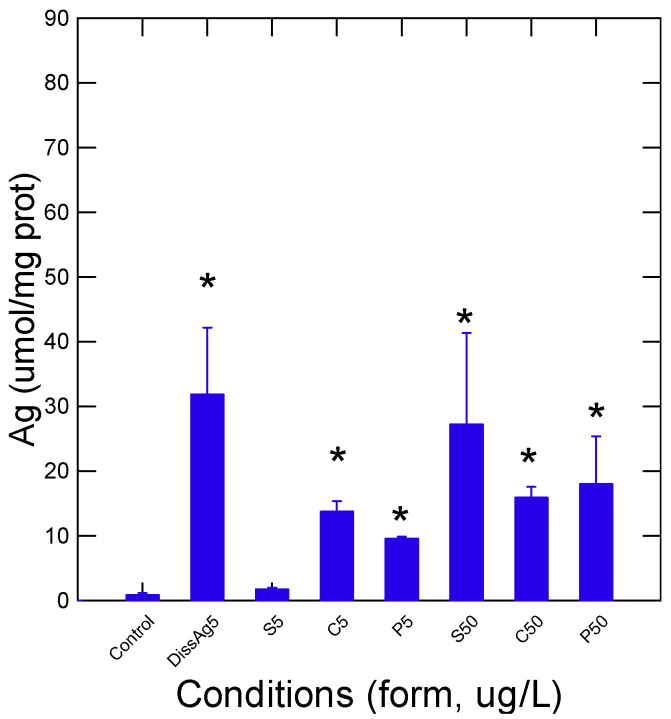
Relative Ag levels in fish gills. The relative levels of Ag were determined by an enhanced luminol/Na_2_CO_3_ chemiluminescence in gill homogenates. The data represent the mean with standard error. Abbreviations for x-axis (and for the following figures): DissAg5 (5 µg/L of dissolved Ag), S5 (spherical nAg; 5 µg/L), C5 (cubic nAg; 5 µg/L), P5 (prismatic nAg; 5 µg/L), S50 (spherical; nAg, 50 µg/L). The star * symbol indicates significance at *p* ≤ 0.05 from controls.

**Figure 2 nanomaterials-13-01356-f002:**
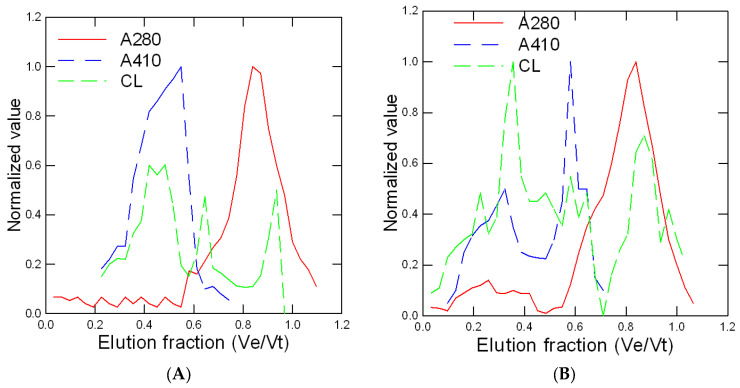
Size-exclusion chromatography analysis of gills in fish exposed to various forms of Ag. The gill homogenate fractions were analyzed by size-exclusion chromatography on a 40 × 1 cm Sephacryl S500 column using 1 mM NaCl and 0.2% Tween-20 as the elution media for spherical (**A**), cubic (**B**), and prismatic (**C**) nAg and dissolved Ag (**D**). The samples were analyzed by a size-exclusion chromatography column pre-calibrated with polystyrene NP (100, 50, and 20 nm diameter), albumin, and NaCl (total volume). The abscissa axis is defined by Ve/Vt, where Ve is the elution volume and Vt is the total column volume (as determined by NaCl). The data were normalized to 1 (fraction signal − min signal)/(max signal − min signal) for ease of measurement for comparisons of each chromatogram.

**Figure 3 nanomaterials-13-01356-f003:**
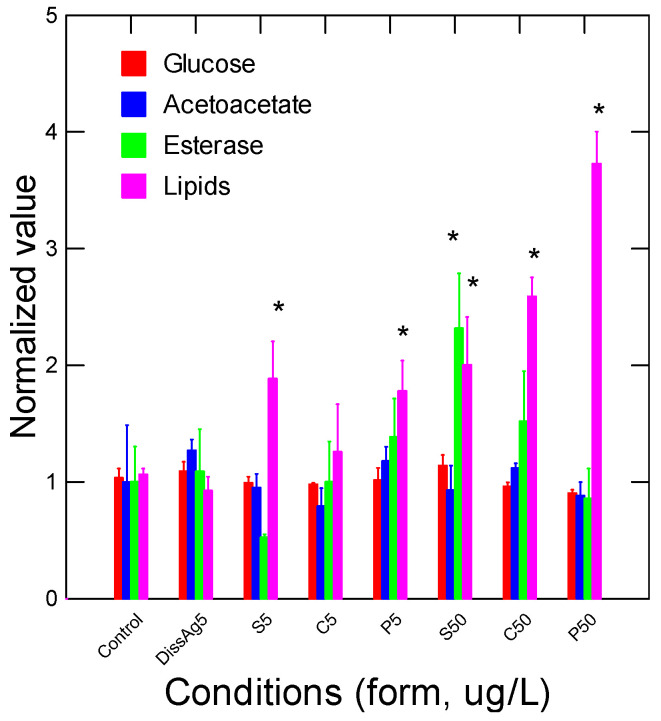
Energy status in trout exposed to nAg forms. Energy status was determined by following the changes in glucose, acetoacetate (a ketone body), esterases, and lipids in trout gills exposed to various forms of Ag. The data represent the mean with standard error. The star * symbol indicates significance at *p* ≤ 0.05 from controls.

**Figure 4 nanomaterials-13-01356-f004:**
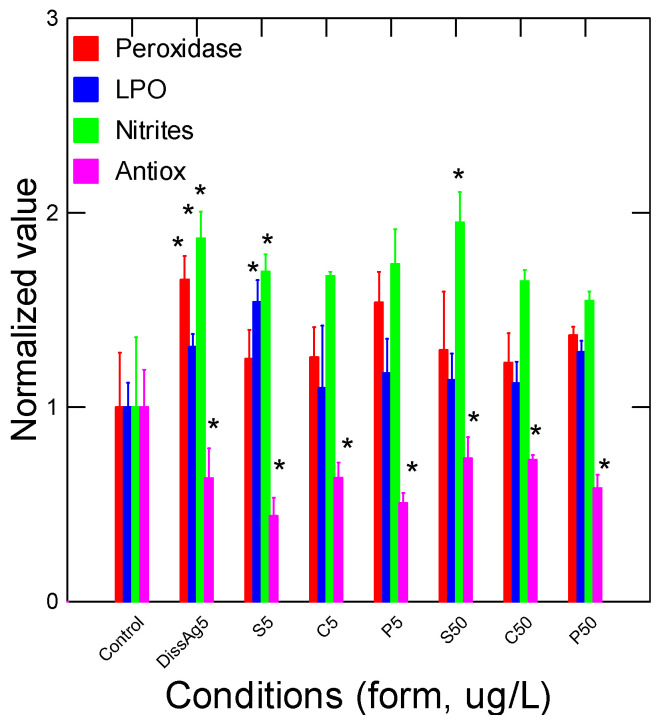
Oxidative stress and damage in trout exposed to nAg forms. Oxidative stress was determined by following the changes in peroxidase, lipid peroxidation, nitrites (NO_2_), and antioxidant (Antiox) potential in gills of rainbow trout exposed to Ag forms. The data represent the mean with standard error. The star * symbol indicates significance at *p* ≤ 0.05 relative to controls.

**Figure 5 nanomaterials-13-01356-f005:**
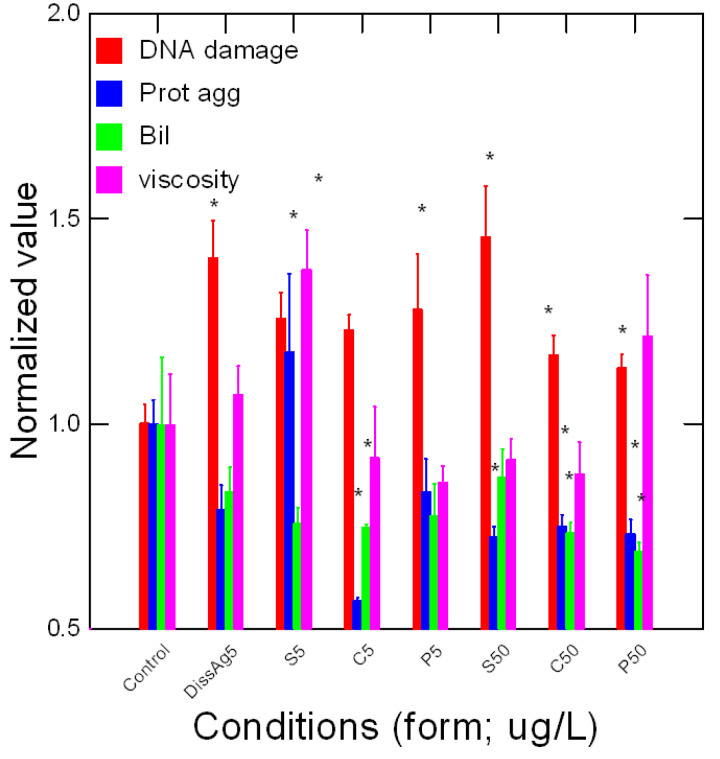
Tissue damage in trout exposed to nAg forms. Tissue damage was evaluated by DNA strand breaks, protein aggregation, and heme degradation (bilirubin) in trout gills exposed to forms of Ag. The data represent the mean with standard error. The star * symbol indicates significance at *p* ≤ 0.05 compared to controls.

**Figure 6 nanomaterials-13-01356-f006:**
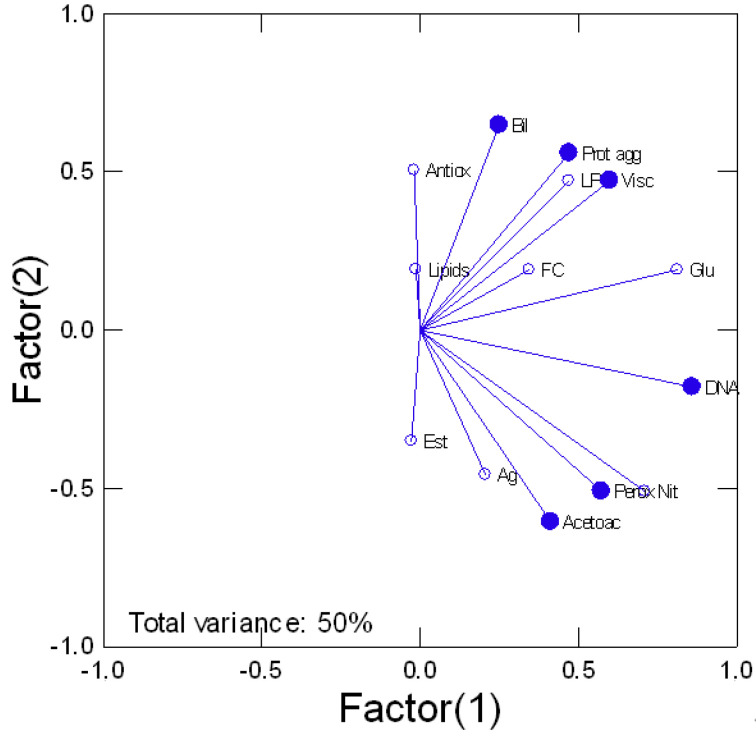
Principal component analysis of biomarker data. The principal component analysis of biomarker data is shown. The total variance explained was at 50% and the most important biomarkers (weight > 0.7) are highlighted with the filled circles. The relative level of Ag is shown as the filled square.

**Table 1 nanomaterials-13-01356-t001:** Stability of Ag in the exposure media.

Ag Form (5 µg/L)	Dissolved Ag 1 h (µg/L)	Dissolved Ag 96 h (µg/L)	Decreasing Rate (h^−1^) /Half Life (h)
Sphere	4.9 ± 0.4	3.1 ± 0.3	0.018/126
Cube	5.2 ± 0.2	2.9 ± 0.2	0.024/114
Prismatic	6.5 ± 0.5	3.4 ± 0.3	0.033/150
Dissolved	4.7 ± 0.4	4.9 ± 0.4	~0

**Table 2 nanomaterials-13-01356-t002:** Change in fish morphology following exposure to the various forms of Ag.

Ag	CF	HIS	GI
Controls	1.02 ± 0.01	0.98 ± 0.07	0.05 ± 0.02
Dissolved	1 ± 0.01	1.19 ± 0.07 *	0045 ± 0.017
Sphere			
5	0.98 ± 0.02	0.9 ± 0.07 *	0.048 ± 0.017
50	0.97 ± 0.01	1.3 ± 0.08 *	0.052 ± 0.018
Cube			
5	0.93 ± 0.009	0.93 ± 0.05	0.05 ± 0.018
50	1.09 ± 0.03	1.2 ± 0.1	0.048 ± 0.017
Prism			
5	1 ± 0.02	0.76 ± 0.02 *	0.051 ± 0.02
50	1 ± 0.02	1.1 ± 0.09	0.054 ± 0.019

The star symbol * indicates significance from controls. CF: condition factor (wet/fork length); HIS: hepatic-somatic index (liver weight/fish weight), GI: gill index (gill weight/fish weight.

## Data Availability

Data are available upon request to the corresponding author.

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
