# Peer review of "Form-Dependent Toxicity of Silver Nanomaterials in Rainbow Trout Gills"

_nanomaterials, 2023, doi:10.3390/nano13081356_

Round 1

Reviewer 1 Report

This is an interesting manuscript and well-written. It can be published before some revision.

1. Authors need to pay attention to the accuracy of the subscript of the chemical formula.

2. The -1 in the full text should be superscripted.

3. There should be spaces between numbers and symbols.

Author Response

Reviewers comments Nanomaterials

All the answers to the comments here and in the text are highlighted in red for easy tracking of the changes.

Rev1

This is an interesting manuscript and well-written. It can be published before some revision.

  1. Authors need to pay attention to the accuracy of the subscript of the chemical formula. Done
  2. The -1 in the full text should be superscripted. Done
  3. There should be spaces between numbers and symbols. Done

Rev2

The manuscript is devoted to the effect of commercial silver nanoparticles of different morphology (sphere, cube, prism) on trout fry. In general, an interesting manuscript on toxicology, however, in my opinion, the manuscript does not fit the journal. The nanoparticles are not characterized in any way in the manuscript, it is not at all clear what kind of nanoparticles we are dealing with. The characterization of the nanomaterials were provided in a previous study. This was added in the text at lines 93-97.

This journal mainly publishes articles devoted to the manufacture and application of new nanoscale objects. I encourage authors to submit their manuscript to the journal Toxics (MDPI) (IF=4,5), where the manuscript will be enthusiastically received!

Rev3

  1. Abstract. Please add more key experimental data. Done, see lines 16-20.
  2. Line 20. Please check the subscript of ‘NO2’. Modifications are required throughout the text. Done
  3. Introduction. Please emphasize the novelty of this manuscript. See lines 74-76.
  4. Line 89. ‘280 uScm-1’ should be revised. Done
  5. Line 128. Please change ‘10°/min’ to ‘10 °C min-1’. I do not see this annotation in the text.
  6. Discussion. The impact of nanoparticles on humans through the food chain should be discussed. We added one sentence for this point although this study focuses on fish impacts and we are far from human consumption. See lines 285-87.
  7. Please add a sentence on the future application prospects in the conclusions. Done see line 371-373.

Reviewer 2 Report

The manuscript is devoted to the effect of commercial silver nanoparticles of different morphology (sphere, cube, prism) on trout fry. In general, an interesting manuscript on toxicology, however, in my opinion, the manuscript does not fit the journal. The nanoparticles are not characterized in any way in the manuscript, it is not at all clear what kind of nanoparticles we are dealing with. This journal mainly publishes articles devoted to the manufacture and application of new nanoscale objects. I encourage authors to submit their manuscript to the journal Toxics (MDPI) (IF=4,5), where the manuscript will be enthusiastically received!

Author Response

(The authors gave the same response as above.)

Reviewer 3 Report

1.     Abstract. Please add more key experimental data.

2.     Line 20. Please check the subscript of ‘NO2’. Modifications are required throughout the text.

3.     Introduction. Please emphasize the novelty of this manuscript.

4.     Line 89. ‘280 uScm-1’ should be revised.

5.     Line 128. Please change ‘10°/min’ to ‘10 °C min-1’.

6.     Discussion. The impact of nanoparticles on humans through the food chain should be discussed.

7.     Please add a sentence on the future application prospects in the conclusions.

Author Response

(The authors gave the same response as above.)

Reviewer 4 Report

The purpose of this study was therefore to compare the toxic effects of various forms of PVP coated nAg of similar dimensions in juvenile rainbow trout in the attempt to address the null hypothesis of this study: the geometry of nAg have no influence towards toxicity. However, there are many research papers related to the biological safety of Ag (e.g., Aquatic Toxicology, 2010, 96(1): 44-52; Comparative Clinical Pathology, 2015, 24: 995-1007.). Therefore, I believe that the innovation of this paper is very limited and is not suitable for publication on Nanomaterials.

1.      The authors did not agree on the form of the chemical formula.

2.      The authors should add a relationship between surface area of different shapes and dissolved silver.

3.      The author's data are simple and unconvincing, so it is suggested to add physical pictures and biological data about animals.

Author Response

The purpose of this study was therefore to compare the toxic effects of various forms of PVP coated nAg of similar dimensions in juvenile rainbow trout in the attempt to address the null hypothesis of this study: the geometry of nAg have no influence towards toxicity. However, there are many research papers related to the biological safety of Ag (e.g., Aquatic Toxicology, 2010, 96(1): 44-52; Comparative Clinical Pathology, 2015, 24: 995-1007.). Therefore, I believe that the innovation of this paper is very limited and is not suitable for publication on Nanomaterials.

  1. The authors did not agree on the form of the chemical formula. This was corrected in the revision.
  2. The authors should add a relationship between surface area of different shapes and dissolved silver. This was added in the text at lines 99-101.
  3. The author's data are simple and unconvincing, so it is suggested to add physical pictures and biological data about animals. The data were significantly changed in some cases but we added a table (table 2) on morphometrics as per the suggestion. See lines 215-18.

Reviewer 5 Report

Review on the manuscript of Auclair J et al.: “Form dependent toxicity of silver nanomaterials in rainbow trout.”.

In this manuscript, the authors explored the potential toxicity of silver nanoparticles to rainbow trout. Authors observed a higher accumulation of dissolved Ag in fish gills than spheric, cubic and prismatic Ag nanoparticles, which translates in higher protein aggregation. In addition, Ag nanoparticles and dissolved Ag induced an increase in the amount of lipids, peroxidase activity, DNA breaks and lipid peroxidation and nitrite levels, and reduced the antioxidant capacity in the gills.   

Overall, I felt that the study has a rational to be performed. Thus, the issues that arise to me are listed below for consideration of the authors.

1 - I recommend the authors to carefully read the manuscript and correct some grammatical mistakes and errors in phrase construction like the following “Moreover, detectable amounts of proteins (A280 nm) were present on prismatic nAg suggesting”.

2 - Authors reported that at low concentrations, cubic and prismatic nAg accumulated more in gills compared to spheric nAg. Do the authors have an explanation for that? If so, this information could be included in the discussion section.

3 - In figure 2, panels A and B are not identified.

4 - In figure 3, the normalized values for glucose and Lipids’ levels in control conditions is higher than 1, but it should be equal to 1. The same happens for protein aggregation data shown in figure 5 (control condition). It suggests some mistakes in data analysis. So, I recommend authors to revisit the analysis of the data to be sure about that.

5 - I recommend the authors to cite figure 5 while describing the data, as there is no mention to figure 5 in the text.

Author Response

Review on the manuscript of Auclair J et al.: “Form dependent toxicity of silver nanomaterials in rainbow trout.”.

In this manuscript, the authors explored the potential toxicity of silver nanoparticles to rainbow trout. Authors observed a higher accumulation of dissolved Ag in fish gills than spheric, cubic and prismatic Ag nanoparticles, which translates in higher protein aggregation. In addition, Ag nanoparticles and dissolved Ag induced an increase in the amount of lipids, peroxidase activity, DNA breaks and lipid peroxidation and nitrite levels, and reduced the antioxidant capacity in the gills.   

Overall, I felt that the study has a rational to be performed. Thus, the issues that arise to me are listed below for consideration of the authors.

1 - I recommend the authors to carefully read the manuscript and correct some grammatical mistakes and errors in phrase construction like the following “Moreover, detectable amounts of proteins (A280 nm) were present on prismatic nAg suggesting”. The manuscript was reread for grammar and this sentence was changed.

2 - Authors reported that at low concentrations, cubic and prismatic nAg accumulated more in gills compared to spheric nAg. Do the authors have an explanation for that? If so, this information could be included in the discussion section. Although we did observe this, an explanation for this eludes us at present. A possible for this is that the cubic and prismatic nAg would release more dissolved Ag and react more strongly with the chemoluminescent assay (in other words, it is a results of more reactive Ag).See lines 282-85.

3 - In figure 2, panels A and B are not identified. Done.

4 - In figure 3, the normalized values for glucose and Lipids’ levels in control conditions is higher than 1, but it should be equal to 1. The same happens for protein aggregation data shown in figure 5 (control condition). It suggests some mistakes in data analysis. So, I recommend authors to revisit the analysis of the data to be sure about that. This was corrected.

5 - I recommend the authors to cite figure 5 while describing the data, as there is no mention to figure 5 in the text. Corrected

Round 2

Reviewer 2 Report

I had no serious comments on the manuscript. If all other reviewers think the manuscript is a good fit, so be it.

Reviewer 3 Report

OK

Reviewer 4 Report

There are many research papers related to the biological safety of Ag (e.g., Aquatic Toxicology, 2010, 96(1): 44-52; Comparative Clinical Pathology, 2015, 24: 995-1007.). Therefore, I believe that the innovation of this paper is very limited and is not suitable for publication on Nanomaterials.